# Retrospective Analysis of the Airway Space Changes in Dentofacial Deformity after Two-Jaw Orthognathic Surgery Using Cone Beam Computed Tomography

**DOI:** 10.3390/jpm13081256

**Published:** 2023-08-14

**Authors:** Víctor Ravelo, Gabriela Olate, Alejandro Unibazo, Márcio de Moraes, Sergio Olate

**Affiliations:** 1Grupo de Investigación de Pregrado en Odontología (GIPO), Universidad Autónoma de Chile, Temuco 4810101, Chile; 2Center for Morphological and Surgical Studies (CEMyQ), Universidad de La Frontera, Temuco 4811230, Chile; 3Department of Oral and Maxillofacial Surgery, AGP Hospital, Lautaro 4860133, Chile; 4Division of Oral and Maxillofacial Surgery, Piracicaba Dental School, State University of Campinas, Piracicaba 13414-903, SP, Brazil; 5Division of Oral, Facial and Maxillofacial Surgery, Universidad de La Frontera, Temuco 4811230, Chile

**Keywords:** airway, orthognathic surgery, OSAS

## Abstract

Orthognathic surgery is used to modify anomalies in maxillomandibular position; this process can significantly affect the anatomy of the airway and cause functional changes. This study aims to define the impact of mandibular maxillary movement on the airway of subjects with dentofacial deformity. A retrospective study was conducted on subjects with Angle class II (CII group) and Angle class III (CIII group) dentofacial deformities. The subjects were treated via bimaxillary surgery; for all of them, planning was performed with software and 3D printing. Cone beam computed tomography (CBCT) was obtained 21 days before surgery and 6 months after surgery and was used for planning and follow-up with the same conditions and equipment. Was used the superimposition technique to obtain the maximum and minimum airway areas and total airway volume. The data were analyzed with the Shapiro–Wilk test and Student’s *t*-test, while Spearman’s test was used to correlate the variables, considering a value of *p* < 0.05. Thus, 76 subjects aged 18 to 55 years (32.38 ± 10.91) were included: 46 subjects were in CII group, treated with a maxillo-mandibular advancement, and 30 subjects were in the CIII group, treated with a maxillary advancement and a mandibular setback. In the CII group, a maxillary advancement of +2.45 mm (±0.88) and a mandibular advancement of +4.25 mm (±1.25) were observed, with a significant increase in all the airway records. In the CIII group, a maxillary advancement of +3.42 mm (±1.25) and a mandibular setback of −3.62 mm (±1.18) were noted, with no significant changes in the variables measured for the airway (*p* > 0.05). It may be concluded that maxillo-mandibular advancement is an effective procedure to augment the airway area and volume in the CII group. On the other hand, in subjects with mandibular prognathism and Angle class III operated with the maxillary advancement and mandibular setback lower than 4 mm, it is possible to not reduce the areas and volume in the airway.

## 1. Introduction

The maxillo-mandibular structures are modified by the development and growth in three dimensions and can be related to anatomical changes, such as the volume and shape of the airway [1]. Changes in the middle and lower third of the face with clockwise or counterclockwise rotation of the mandible are related to the position of the hyoid bone, the morphology of the facial thirds and the pharyngeal spaces [2,3].

Craniofacial deformity and obesity are important etiological factors for developing obstructive sleep apnea syndrome (OSAS) [4,5,6]. OSAS can be treated using different techniques, and orthognathic surgery is one of them. The surgery to modify the position of the maxillary and mandibular bones is related to anatomical changes in the functional morphology and significant changes in airway volume, as observed in maxillo-mandibular advancement (MMA) [7]. It has been suggested that changes due to MMA improve anatomical conditions, sleep quality and oxygen saturation [8]. 

One of the aims of orthognathic surgery is to achieve a suitable orofacial functional and facial harmony. The current trend in orthognathic surgery is the forward movement of the facial structures [9], as a chance to create positive changes in esthetics and the pharyngeal spaces [10]. Bimaxillary advancement is common in subjects with a class II dentofacial deformity; however, in subjects with a class III dentofacial deformity, it is still common to observe mandibular setback surgeries in conjunction with maxillary advancement. Thus, the impact on the airway could be negative in subjects with class III deformities who undergo mandibular setback surgeries.

This study aims to analyze the airway morphology after orthognathic surgery in subjects with a class II or class III dentofacial deformity and the impact on the airway space.

## 2. Materials and Methods

A retrospective study was conducted to evaluate the airway condition after orthognathic surgery in patients with retrognathism or prognathism. Inclusion criteria were for subjects between 18 and 45 years old of both sexes with an Angle type II or III and who underwent maxillary and mandibular orthognathic surgery; the ANB angle was used to define dental discrepancy and to confirm the dentofacial deformity, and none of the patients had respiratory dysfunction requirements. Subjects with previous orthognathic surgery, medical records with facial trauma, congenital syndrome or malformations and subjects with facial asymmetry with a chin deviation greater than 5 mm from the facial midline were excluded. The patients included in this research provided written informed consent. The Declaration of Helsinki was respected in this research.

### 2.1. Orthognathic Surgery

The surgical planning was conducted with the digital workflow, and the surgery was performed using a 3D-printed surgical splint. All the osteotomies were carried out with a piezoelectric system (Satelec, Action, France). The Le Fort I type non-segmented osteotomy (LFI) was used as regular with intraoral approach and fixation with four 2.0 osteosynthesis plates and monocortical screw (Enterprises, Artfix Implants, Pinhais, PR, Brazil) [11]. The bilateral sagittal split ramus osteotomy (BSSO) was performed with intraoral approach and the use one or two 2.0 miniplates with monocortical screws (Enterprises, Artfix Implants, Pinhais, PR, Brazil). No other osteotomy, as genioplasty, was performed in the patients, and no other complementary surgery, such as as non-invasive or invasive temporomandibular joint (TMJ) treatment, cosmetic surgery or bone grafts, was included

The main indication for orthognathic surgery was malocclusion, dysfunction in stomatognathic area and aesthetic complaints. Exams for respiratory function or sleep apnea were not included because in this clinical series, none of the patients had the requirement in airway augmentation. 

The mandible-first approach was used in all patients; the surgical movements were selected based on the individual requirement according to dental occlusion, cephalometry and facial analysis, considering the esthetic requirements of each patient. A bimaxillary advancement was designed for patients with Angle class II; for those with Angle class III, a maxillary advancement and mandibular setback were designed.

### 2.2. Image Analysis

The image was captured with the NewTom 3D VGi EVO CBCT device (Verona, Italy), the field of view 24 × 19 cm and exposure parameters 110 kV, 8 mA and 15 s. A trained expert technician who specializes in imaging took the image. The patient was placed in a still, vertical position, with the lips at rest and without forcing a muscle position. Once the image was obtained, the New Tom NNT software was used (Imola, Italy). The preoperative imaging studies were conducted within 21 days before the surgery, and the postoperative checkups were performed between 5 and 6 months after the surgery (Figure 1 and Figure 2). 

### 2.3. Airway Analysis

An algorithm was created in the specific software to establish the total airway volume and minimum and maximum area. 

The landmarks used in this research were: -Anterior: posterior nasal spine in the sagittal plane and choanae in the axial plane.-Posterior: posterior wall of the pharynx.-Superior: highest point of the nasopharynx.-Inferior: under hyoid bone, at the level of the lower edge of the C4 vertebral body.

### 2.4. Superimposition Analysis of Maxillo-Mandibular Movement

For the overlap process, Nasion (N)—Sella (S)—Porion (Po) and the zygomaticomaxillary suture (Z) were used as fixed points. These points were overlapped on a preoperative and postoperative CBCT, and the movement of the anterior nasal spine (ANS), Point A, Point B, Chin (Me) and hyoid bone (H) was assessed (Figure 1 and Figure 2).

### 2.5. Statistical Analysis

Measurements were taken in 20 studies by the same observer at a two-week interval. An intraclass index of 0.81 for continuous variables was obtained. The data analysis was performed with Graph Prism v. 9.5.1. The clinical parameters are presented as mean (X) and standard deviation (SD). The Shapiro–Wilk test was used for the analysis of normal distribution. To evaluate and compare the continuous variables of the overlap, a Student’s t-test was used. Spearman’s test was used for the correlation between the variables, considering a value of *p* < 0.05 as a significant difference.

## 3. Results

Seventy-six subjects were included, with an age between 18 and 55 years (32.38 ± 10.91). Twenty-six were male (34.2%), and fifty were female (65.8%). Forty-six subjects received a maxillo-mandibular advancement, and thirty had a maxillary advancement and mandibular setback. 

In the case of the CII group (Table 1), after the MMA, significant changes were observed in the increase in minimum airway area (*p* < 0.0001), maximum airway area (*p* < 0.006) and total volume (*p* < 0.0001). The minimum airway area increased by 28.14 mm^2^ (±26.22), the maximum by 80.62 mm^2^ (±23.66) and the total volume by 8.4 cm^3^ (±5.11) (Table 1). In the case of the CIII group (Table 2), after the maxillary advancement and mandibular setback, differences of 1.37 mm^2^ (±1.59) were noted in the minimum airway area, 18.32 mm^2^ (±3.37) in the maximum area and 2.04 cm^3^ (±1.88) in the total volume; the changes in this group were not significant in any airway measurement. 

Regarding surgical movement (Table 3), in the CII group, a maxillary advancement of +2.45 mm (±0.88) and a mandibular advancement of +4.25 mm (±1.25) were observed. In the CIII group, a maxillary advancement of +3.42 mm (±1.25) and a mandibular setback of −3.62 mm (±1.18) were observed. The surgical movement performed in both groups was significant (*p* < 0.001) for each group and each maxillary movement. 

It is interesting to note that the maxillary advancement of the CII group was almost 1 mm less than the same surgical movement in the subjects in the CIII group and that this difference was not statistically significant (*p* < 0.1); in the case of the mandible, the difference between the CII group and CIII group was almost 8 mm, and this difference was significant. On other hand, the measurement in group III was in augmentation in the postoperative analysis with no statistical differences. 

## 4. Discussion

The method used in this research has been used in the past. The CBCT images from the diagnosis and follow-up were used in superimposition with surface-based methods in the voxel and reference points [12] to perform comparison. This makes it easier to understand the clinical results and the stability of surgical movement [13]. In our study, we used the superimposition method with the base of the skull as a reference to evaluate the displacement of the maxilla and mandible, which allows one to confirm the real movement obtained after surgery.

Maxillo-mandibular advancement surgery in subjects with type II dentofacial deformity can produce augmentation in anatomical conditions in the oropharyngeal space when comparing to subjects with a type III dentofacial deformity treated with maxillary advancement and mandibular setback [14]. Several authors [15,16] indicated that the morphological changes in the airway of subjects with type III dentofacial deformity were negative after immediate postoperative evaluation of mandibular setback. However, after 6 months, there were no significant long-term changes, being stable. For that reason, our research includes the 6-month follow-up in the CII and CII groups. On the other hand, although there were retrognathic subjects with MMA, there is also a slight tendency for the airway to be reduced after 6 months, and this change does not generate complications or reductions that lead to a collapse of the airways [17].

The MMA in subjects with mandibular retrognathia showed a significant increase in airway volume, generating important benefits in the treatment of the OSAS, regardless of sex or ethnic group [18,19,20]. Our results agree with several MMA studies, where subjects with type II dentofacial deformity showed significant changes in maximum and minimum airway areas and total airway volume. We also observed a slight increase in airway volume in type III dentofacial deformity subjects who underwent a maxillary advancement and mandibular setback with no statistical differences. This point is interesting because the bimaxillary movement with maxillary advancement greater than 3 mm could justify the increased airway and prevent the mandibular setback from having a significant impact.

Khaghaninejad et al. [21] evaluated the changes in the pharyngeal airway in 48 subjects with type III dentofacial deformity, comparing groups who received only mandibular setback surgery (G1), combined maxillary advancement and mandibular setback surgery (G2) and maxillary advancement surgery (G3). It was noted that the G1 procedures generated the greatest reduction in volume, followed by G2 and G3. On the other hand, An et al. [15] performed a 6-year follow-up of mandibular prognathism subjects who underwent orthognathic surgery, noting that a maxillary advancement between 1.90 ± 1.31 mm and a mandibular setback of 8.14 ± 4.62 mm produced a slight reduction in oropharyngeal volume 6 months postoperatively, but in the follow-up until 6 years, there were no significant changes. 

Our work showed that, in retrognathic subjects with MMA movements of 2.45 ± 0.88 mm in the maxilla and 4.25 ± 1.25 mm in the mandible, significant changes are achieved in the increased maximum, minimum and total airway volume. However, in prognathic subjects who underwent maxillary advancements of 3.42 ± 1.25 mm and mandibular setbacks of 3.62 ± 1.18, there was a slight increase in airway volume with no statistical differences. The use of maxillary advancements greater than 3 mm in CIII type dentofacial deformity included some objectives, such as facial balance, stable dental occlusion, reduction in the mandibular setback requirements and to obtain an stable airway. 

The airway volume obtained in the CIII group could be associated with a compensatory trend of the maxillary advancement and the lower movement in mandibular setback, decreasing the chance of collapse of the soft tissue in the airway. Yang et al. [22], using computed tomography and polysomnography as diagnostic methods, demonstrated that when performing bimaxillary orthognathic surgery with a mandibular setback greater than or equal to 9 mm on 12 subjects without OSAS, the pharyngeal, oropharyngeal and hypopharyngeal volumes were significantly reduced. Four of the twelve subjects developed mild OSAS 6 months after the surgery. Mandibular setback movements lower than 4 mm could not be related to a significant reduction in airway volume in our sample. 

Facial deformities with an angle class II and angle class III malocclusion are progressive deformities. The presence of a class II skeletal condition can be related to deficiency in airway volume and can be associated with some comorbidities, such as sleepiness and cognitive deficiency, and in the adolescence group, would be particularly complex because of the physical and psychological growth [23]. In the same line, adolescents with OSAS show a high risk of major cardiovascular diseases [24]. Anatomical characteristics show implications in OSAS development [23] and, for that reason, one of the aims in orthognathic surgery is to obtain better conditions in the anatomical volume of the airway or at least to maintain the morphology if no augmentation is necessary. 

In this research, sleep disorder was not included in the diagnosis; however, the airway was the main area of analysis because of the potential involvement in sleep-disordered breathing and their role for another physical disorder. The effect of sleep disorders in cardiovascular health, the respiratory system and neurocognitive conditions has been studied for a long time [25]. Orthognathic surgery can help to treat patients under these conditions; however, non-surgical management is the initial step. 

The weight loss decreases the effect of the sleep disorder; some research confirmed that 10% in weight augmentation can be related to a 32% increase the apnea–hypopnea index [25]; in the same line, the diet and habits, like the use of alcohol and smoking, can influence sleep disorders. It has been demonstrated that a change in some habits with a healthy life will help in the first line of treatment [26,27]. 

Dental management with oral devices such as mandibular repositioning devices can be used to move the mandibular position in a more anterior position and to achieve the movement of the tongue and oropharynx area and to create a change in the airway with temporal augmentation [28]. This system can be used in patients with mild to moderate sleep disorders and can be effective in some patients. However, oral devices can be less effective than CPAP in the treatment of obstructive sleep apnea [28]; on other hand, the requirements for this type of device are low, being realized with a low cost and fast to use with easy management for clinicians and patients. Disadvantages of the mandibular repositioning device include the dental pain, tmj pain, changes and dental occlusion, periodontal disease and other dental pathologies. Patient compliance is over 50%, showing a moderate adherence [28].

The use of continuous positive airway pressure (CPAP) and bilevel positive airway pressure (BiPAP) is a confirmed strategy to treat sleep disorders. The proper use of CPAP can eliminate sleepiness and reduce or eliminate hypertension; use for more than 18 months can help to maintain a better quality of life [29]. Although CPAP is an effective tool to treat sleep disorders, its acceptability by patients can be low in some cases; the noise, difficulties to use in a trip, electricity requirements, size and position of the machine and the requirement for a good fit cannot be tolerated by some patients. Noncompliance in the use of CPAP can be over 50% [30].

Surgical techniques are included in second-line treatment [31]. Phase I includes nasal surgery, uvulopalatopharyngoplasty (UPPP) and genioglossus/hyoid reposition, and phase II includes bimaxillary advancement or tongue reduction. In class III dentofacial deformity, the maxilla-mandibular advancement can be difficult to obtain because of the previous mandibular position and can be a challenge for surgeons in terms of planning the surgery. The maxilla-mandibular movement has some limits, so the proper occlusion has to be obtained in all scenarios.

The airway can be influenced by the anatomical condition of the subjects, including weight, position of tongue, presence of tonsils, hyoid bone anatomy and other variables [32], and the air flow can be related to stress, the abuse of alcohol, sleep habits and other human conduct [33], so it is difficult to define sleep disorders based on the anatomical airway morphology and the change included in our research; however, our results show that using bimaxillary orthognathic surgery can be an augmentation of the airway in retrognathic subjects and no change in the case of prognathic subjects treated using maxillary advancement and mandibular setback.

Limitations in this research can be the low sample, the short time for follow-up of the patients, the variability in diagnosis and the absence of a sleep disorder test in the preoperative and postoperative analysis. However, this research shows a clear route to perform a good planning of orthognathic surgery, looking to maintain a well-functioning airway, as well as a stable facial balance and proper dental occlusion.

## 5. Conclusions

At the 6-month follow-up, we can conclude that maxilla-mandibular advancement is an effective treatment to augmentation of the airway area and volume retrognathic subjects. On the other hand, in subjects with mandibular prognathism and Angle class III operated with a maxillary advancement and mandibular setback lower than 4 mm, it is not possible to reduce the area and volume in the airway.

## Figures and Tables

**Figure 1 jpm-13-01256-f001:**
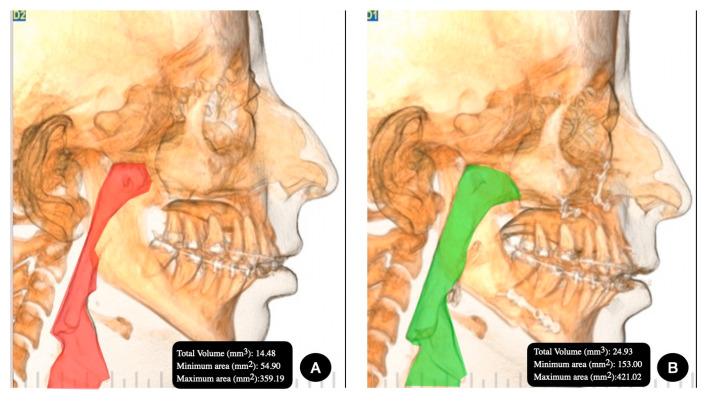
Group II, Angle class II facial deformity in the (**A**) preoperative image and (**B**) postoperative image.

**Figure 2 jpm-13-01256-f002:**
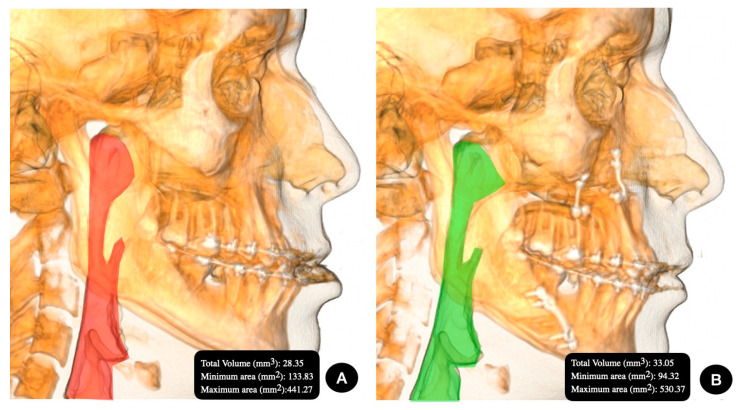
Group III, Angle class III facial deformity in the (**A**) preoperative image and (**B**) postoperative image.

**Table 1 jpm-13-01256-t001:** Minimum and maximum airway area and total airway volume in relation to the preoperative and postoperative skeletal position in subjects with Angle class II facial deformity. The average maxillary advancement was +2.45 mm (±0.88), and the average mandibular advancement was +4.25 mm (±1.25).

	Minimum Area (mm^2^)			Maximum Area (mm^2^)			Total Volume (cm^3^)		
	X	SD	*p* Value	X	SD	*p* Value	X	SD	*p* Value
Pre Op	125.6	119.29	0.0001 *	474.72	142.05	0.006 *	24.79	6.69	0.0001 *
Post Op	153.74	92.92		555.34	118.35		33.19	11.80	
Difference	(+) 28.14	±26.22		(+) 80.62	±23.66		(+) 8.4	±5.11	

Note: X: average of measurements; SD: standard deviation. (*) indicates a statistically significant difference. Surgical movement includes bimaxillary advancement.

**Table 2 jpm-13-01256-t002:** Minimum and maximum airway area and total airway volume in relation to the preoperative and postoperative skeletal position in subjects with Angle class III facial deformity. The average maxillary advancement was +3.42 mm (±1.25), and the average mandibular setback was −3.62 mm (±1.18).

	Minimum Area (mm^2^)			Maximum Area (mm^2^)			Total Volume (cm^3^)		
	X	SD	*p* Value	X	SD	*p* Value	X	SD	*p* Value
Pre Op	142.67	66.55	0.91	561.13	93.69	0.2	32.67	8.73	0.11
Post Op	144.04	84.09		585.45	157.06		34.71	16.16	
Difference	(+) 1.37	±1.59		(+) 18.32	±3.37		(+) 2.04	±1.88	

Note: Pre Op: preoperative; Post Op: postoperative; X: average of measurements; SD: standard deviation. Surgical movement includes maxillary advancement and mandibular setback. (+): positive difference in the post op.

**Table 3 jpm-13-01256-t003:** Minimum and maximum airway space and superposition by preoperative and postoperative image of bimaxillary movement in subjects with Angle class II and III facial deformity.

	Superposition T1–T2				
	CII Group (mm)		CIII Group (mm)		*p* Value
	X	SD	X	SD	
Maxillary movement	(+) 2.45	0.88	(+) 3.42	1.25	0.3
Mandibular movement	(+) 4.25	1.25	(−) 3.62	1.18	0.1
Minimum area	(+) 28.14	26.22	(+) 1.37	1.59	0.0001 *
Maximum area	(+) 80.62	23.66	(+) 18.32	3.37	0.0001 *
Total volume	(+) 8.4	5.11	(+) 2.04	1.88	0.0001 *

Note: T1: preoperative; T2: postoperative; CII: Angle class II facial deformity; CIII: Angle class III facial deformity; Positive (+): advancement movement; Negative (−): setback movement; X: average of measurements; SD: standard deviation. (*) indicates a statistically significant difference.

## Data Availability

The data are available upon request from the corresponding author.

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
