# Peer review of "Retrospective Analysis of the Airway Space Changes in Dentofacial Deformity after Two-Jaw Orthognathic Surgery Using Cone Beam Computed Tomography"

_jpm, 2023, doi:10.3390/jpm13081256_

Round 1
Reviewer 1 Report
This article is well-prepared, and the topic is interesting. I have some remarks.
First of all, the Ethical committee approval needs to be included.
Second, some information about patients included in this study should be added. It is not clear which were indications for orthognathic surgery. Was it malocclusion alone, or had some patients also OSAS? Were all surgeries LF 1 osteotomy and BSSO (this abbreviation is used more often than BSSRO) only? Or did some segmental osteotomies or genioplasties also take part in the surgery?
Author Response
- This article is well-prepared, and the topic is interesting. I have some remarks.
R. Thank you
- First of all, the Ethical committee approval needs to be included.
R. this was included in the las statement of the article and in the M&M
- Second, some information about patients included in this study should be added. It is not clear which were indications for orthognathic surgery. Was it malocclusion alone, or had some patients also OSAS? Were all surgeries LF 1 osteotomy and BSSO (this abbreviation is used more often than BSSRO) only? Or did some segmental osteotomies or genioplasties also take part in the surgery?
R. new info was included to get better info.
Reviewer 2 Report
Airway Space Changes in Dentofacial Deformity After Two-Jaw 2 Orthognathic Surgery in Class II and Class III Skeletal Conditions
The paper is interesting and well written
At the 6-month follow-up, we can conclude that maxillo mandibular advancement is an effective treatment to augmentation of the airway area and volume retrognathic subjects. On the other hand, in subjects with mandibular prognathism and Angle class III operated with a maxillary advancement and mandibular setback lower than 4 mm is possible to not reduce the area and volume in the airway
The title does not adequately reflect the content of the paper.
The title should include the studio's design and also Cone beam computed tomography
quantification of pharyngeal airway space…
Introduction Should situate the problem of the study
Should define inclusion and exclusion criteria more clearly. surgery in patients with retrognathism or prognathism
The following has not been obtained The airway can be influenced by the anatomical condition of the subjects as weight Body mass index (BMI), position of tongue, presence of tonsils, hyoid bone anatomy and others variables to stress, the abuse of alcohol, sleep habits and other human
Indicate limitations and weaknesses and strengths
Author Response
1. The paper is interesting and well written
R. Thank you
2. The title does not adequately reflect the content of the paper. The title should include the studio's design and also Cone beam computed tomography
R. Title was modified
3. Introduction Should situate the problem of the study
R. introduction was maintained showing the problem as suggested by another reviewer. However, discussion was in an augmentation to include more info about OSAS as suggested by another reviewer as well.
4. Should define inclusion and exclusion criteria more clearly. surgery in patients with retrognathism or prognathism
R. more clear criteria was added in the main text
5. Indicate limitations and weaknesses and strengths
R. this info was added.
Round 2
Reviewer 2 Report
I agree with the chI agree with the changes